# Maintenance of physical activity level, functioning and health after non-pharmacological treatment of pelvic girdle pain with either transcutaneous electrical nerve stimulation or acupuncture: a randomised controlled trial

Annika Svahn Ekdahl [1], Monika Fagevik Olsén,[1,2] Tove Jendman,[3] Annelie Gutke[1]

[1]Department of Health and Rehabilitation; Physiotherapy, Sahlgrenska Academy, Gothenburg, Sweden
[2]Department of Physiotherapy, Sahlgrenska University Hospital, Gothenburg, Sweden
[3]Physiotherapy Clinic 'I Rorelse', Gothenburg, Sweden

**Correspondence to**
Annika Svahn Ekdahl;
annika.svahn-ekdahl@neuro.gu.se

## ABSTRACT

**Objective** To investigate if there are differences between acupuncture and transcutaneous electrical nerve stimulation (TENS) as treatment for pelvic girdle pain (PGP) in pregnancy in order to manage pain and thus maintain health and functioning in daily activities and physical activity (PA).

**Design** Randomised controlled trial.

**Setting and participants** Pregnant women (n=113) with clinically verified PGP in gestational weeks 12–28, recruited from maternity healthcare centres, randomised (1:1) into two groups. Exclusion criteria: any obstetrical complication, systemic disease or previous disorder that could contradict tests or treatment.

**Interventions** The intervention consisted of either 10 acupuncture sessions (two sessions per week) provided by a physiotherapist or daily home-based TENS during 5 weeks.

**Primary outcome variables** Disability (Oswestry Disability Index), functioning (Patient Specific Functional Scale), work ability (Work Ability Index) and PA-level according to general recommendations.

**Secondary outcome variables** Functioning related to PGP (Pelvic Girdle Questionnaire), evening pain intensity (Numeric Rating Scale, NRS), concern about pain (NRS), health (EuroQoL 5-dimension), symptoms of depression/catastrophising (Edinburgh Postnatal Depression Scale/Coping Strategies Questionnaire).

**Results** No mean differences were detected between the groups. Both groups managed to preserve their functioning and PA level at follow-up. This may be due to significantly (p<0.05) reduced within groups evening pain intensity; acupuncture −0.96 (95% CI −1.91 to −0.01; p=0.049), TENS −1.29 (95% CI −2.13 to −0.44; p=0.003) and concern about pain; acupuncture −1.44 (95% CI −2.31 to −0.57; p=0.0012), TENS −1.99 (95% CI −2.81 to −1.17; p<0.0001). The acupuncture group showed an improvement in functioning at follow-up; 0.82 (95% CI 0.01 to 1.63; p=0.048)

**Conclusion** Treating PGP with acupuncture or TENS resulted in maintenance of functioning and physical

### Strengths and limitations of this study

► All participants had clinically verified pelvic girdle pain (PGP) during pregnancy.

► Interventions were evaluated both in the second and third trimester of pregnancy, with good adherence in both intervention groups.

► Lower self-reported satisfaction in the transcutaneous electrical nerve stimulation (TENS) group could be a result of less contact with the treating physiotherapist than in the acupuncture group.

► It is possible that the participants did not use TENS according to instructions, as this was self-reported.

► The study design, where pregnant women had to travel to the clinic for assessment and/or intervention, may have affected the ability to recruit women with more severe PGP to the study.

activity and also less pain and concern about pain. Either intervention could be recommended as a non-pharmacological alternative for pain relief and may enable pregnant women to stay active.

**Trial registration number** 12726. https://www.researchweb.org/is/sverige/project/127261

## INTRODUCTION

Pelvic girdle pain (PGP) and/or low back pain (LBP) is reported internationally by over 50% of pregnant women.[1–3] PGP has musculoskeletal origin and is localised to the area between the posterior iliac crest and the gluteal fold, most commonly around the sacroiliac joints and/or pubic bone, sometimes with radiating pain in the thighs.[4] Limitations in daily functioning and walking difficulties, especially a feeling of 'catching' of the leg are common symptoms.[2] These symptoms differ from those associated with LBP, which is defined as

pain in the area between the lower ribs and the iliac crest, with or without radiating pain to the legs, and is often described as worse when sitting.[5]

The aetiology of PGP is multifactorial. Increased pregnancy-related ligamentous laxity may result in altered mechanics in the pelvic joints.[6] Pain may occur if these alterations cannot be compensated for by neuromuscular control.[4] PGP most commonly occurs around gestation week 18, but can occur at any time during pregnancy. Among women who experience PGP during pregnancy, about 20% suffer from persistent pain 4 months post partum.[7] Previous long-term studies have reported that approximately 10% suffer from remaining PGP more than 11 years post partum.[8 9] Experiencing PGP during pregnancy is a risk factor for having PGP also in a next pregnancy.[2]

Pregnant women are advised to follow the general recommendations for physical activity (PA), of at least 150 min of moderate intensity or at least 75 min of high intensity PA per week or a combination of both.[10] Few pregnant women reach this level of PA and a common cause is PGP.[11] A high level of pain during pregnancy results in more severe limitations in daily life and less PA.[12]

There is a higher rate of sick leave among pregnant women compared with age-matched, non-pregnant women and a major reason is PGP and/or LBP.[13–15] Studies show a reduced risk for sick leave if the woman can continue to stay active and exercise throughout her pregnancy.[16 17] Women with PGP and/or LBP have an increased risk of prenatal anxiety, depressive symptoms and postpartum depression compared with healthy pregnant women.[18 19] If pregnant women maintains the recommended levels of PA, both the odds of getting prenatal depression and the severity of it is reduced.[20] Furthermore, there is a risk in pregnancy for complications such as hypertension, diabetes and pre-eclampsia and this risk is reduced if the woman stays active.[21]

The use of medication in pregnancy has increased over the years[22 23] with a risk for overuse of analgesics among women having PGP.[24] Therefore, it is important to provide efficient treatment alternatives. The best evidence nonpharmacological treatments for PGP are acupuncture, a stabilising pelvic belt and physical exercise.[25 26] There is, however, limited evidence for the use of transcutaneous electrical nerve stimulation (TENS),[25–27] even if it is widely used in physiotherapy clinical practice. The physiological mechanisms of acupuncture and TENS are similar. Both peripheral and central mechanisms of pain control are stimulated, as per the gate control theory, activating descending pathways for pain control and release of endogenous opioids.[28 29] No specific adverse effects regarding use in pregnancy are reported for any of the interventions mentioned here.[30]

The aim of this study was to investigate if there are differences between acupuncture and TENS as treatments for PGP in order to manage pain and thus maintain health and functioning in daily activities and PA.

## METHODS

A randomised, controlled design was used to compare acupuncture and TENS as pain relief for PGP in pregnancy. The decision to randomise into two different groups was made based on ethical considerations. It is not ethical to withhold evidence-based treatment, therefore the experimental treatment (TENS) was given to one group and acupuncture, which is known to be effective for PGP,[25 26] to an active control group.[31]

### Setting and participants

The data collection took place from 2014 to 2018 in two Swedish cities. Two independent physiotherapists located in each city conducted tests and treatment. Women with PGP were briefly informed about the study at maternity healthcare centres and those who were interested were contacted by the independent test leader for detailed information and for initial screening for inclusion.

### Inclusion criteria

► Single fetus pregnancy.
► Pregnancy-related PGP or combined PGP/LBP in gestational weeks 12–28.
► Pain located distally/laterally of L5-S1, over the buttocks and/or the pubic bone, verified by ≥2 positive pain provocation tests or a positive active straight leg raise (ASLR).
► A score of ≥20% on the Oswestry Disability Index (ODI) and/or ≤6 in one self-chosen activity of the Patient-Specific Functional Scale (PSFS).
► Ability to read and understand Swedish.

### Exclusion criteria

► Previous fracture, surgery or malignant disease in the back, pelvis or hips.
► Any systemic disease or obstetrical complication that contraindicates treatment or tests.
► Contraindications for TENS; pacemaker; decreased sensation in the treatment area.
► Contraindications for acupuncture; treatment with anticoagulants.
► Start of other treatment during the study period.

Those eligible for participation underwent a clinical examination to verify PGP and answered the baseline questionnaire. All participants gave oral and written consent. The clinical examination consisted of a reliable procedure, which included a neurological examination, hip rotation range of motion test, pelvic pain provocation tests, a brief mechanical assessment of the lumbar spine and the ASLR test.[32] The examination also included a single-leg balance test.[33]

### Randomisation and interventions

Before start of intervention, the participants each drew a sealed, opaque envelope, prepared from a computer-generated list by an independent research assistant, for assignment to either the acupuncture or TENS group (1:1).

The intervention lasted for 5 weeks in both groups. All women got written general advice on how to manage PGP in addition to the assigned treatment.

## Acupuncture

The intervention consisted of 10 acupuncture sessions (two sessions per week), provided by an experienced physiotherapist, trained in Western medical acupuncture. It was decided, due to ethical considerations, not to continue the intervention for any woman that experienced sufficient pain relief in fewer than 10 sessions. The treating physiotherapist chose the acupuncture points, based on the individual woman's symptoms and clinical presentation, from a protocol published in a previous study where point localisation and insertion depth are described in detail.[34] Disposable, stainless steel needles (China Classic; 0.25×15 mm, 0.30×30 mm, 0.30×50 mm) were used. After insertion and manual stimulation to evoke the sensation of *de Qi* (a radiating sensation or a feeling of heaviness) from the insertion point, the needles were left in situ for 30 min and manually stimulated after 10 and 20 min. The participants got written information about acupuncture at the start of treatment and were instructed to report any perceived side effects to the treating physiotherapist.

## TENS

An experienced physiotherapist tried out the electrode placement according to the individual woman's symptoms and clinical presentation, either unilaterally or bilaterally over the sacroiliac joint and gluteal muscles for dorsal pelvic pain or in the groin area for pubic pain. The women received oral and written instructions regarding their individual electrode placement and then used the TENS device (Cefar Primo PRO; DJO Nordic AB, Malmö, Sweden) at home for at least 30 min per day for 5 weeks. High frequency stimulation (80 Hz) was used with an intensity of the stimulation that produced a strong but comfortable sensation in the treated area. The women were instructed to increase the intensity of the stimulation during the treatment session in order to maintain the strong but comfortable sensation. Each woman met the physiotherapist on a follow-up visit after 1 week to discuss treatment and to adjust electrode placement and stimulation mode if needed. In case of absence of effect after the first week of treatment, the stimulation mode could be changed to low frequency (2 Hz) in order to receive optimal pain relief. After 3 weeks, one follow-up telephone call to each woman was made to ensure that the intervention was working well. After 5 weeks the participants made a final visit to the physiotherapist.

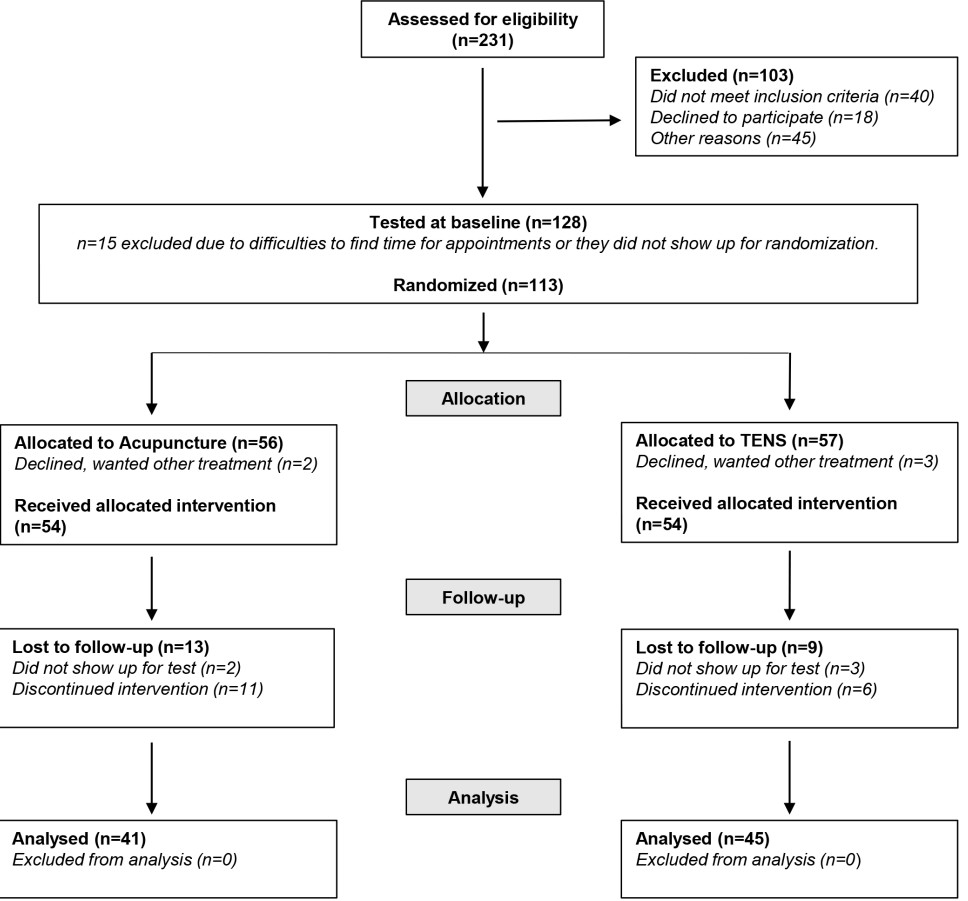

**Figure 1** Flowchart of the study according to the Consolidated Standards of Reporting Trials requirements. TENS, transcutaneous electrical nerve stimulation.

**Table 1** Descriptive data

| Variable | Acupuncture, n=56 | TENS, n=57 | Excluded after baseline test, n=15 |
|---|---|---|---|
| Maternal age, mean (SD), median (min; max) | 30.5 (3.96) 30.0 (20.0; 38.0) | 31.1 (4.38) 32.0 (22.0; 43.0) | 31.9 (3.92) 32.0 (27.0; 40.0) |
| Gestational weeks, mean (SD), median (min; max) | 20.9 (5.32) 21.5 (11.0; 28.0) | 20.8 (5.18) 22.0 (10.0; 28.0) | 22.4 (5.17) 24.0 (13.0; 28.0) |
| BMI, mean (SD), median (min; max) | 25.6 (4.53) 25.0 (19.5; 41.8) | 26.5 (3.87) 26.2 (19.7; 35.0), n=55 | 27.6 (5.83) 27.6 (18.2; 41.0) |
| Parity, n (%) | | | |
| 0 | 21 (37.5) | 22 (38.6) | 6 (40.0) |
| 1 | 30 (53.6) | 28 (49.1) | 5 (33.3) |
| ≥2 | 5 (8.9) | 7 (12.3) | 4 (26.7) |
| Education level, n (%) | | | |
| Elementary school | 1 (1.8) | 2 (3.5) | 0 |
| High school | 12 (22.4) | 20 (35.1) | 3 (20.0) |
| Higher education | 43 (76.8) | 35 (61.4) | 12 (80.0) |
| Occupation, n (%) | | | |
| Paid employment | 47 (83.9) | 47 (82.5) | 14 (93.3) |
| Not in paid employment | 3 (5.4) | 3 (5.3) | 0 |
| Parental leave | 0 | 2 (3.5) | 0 |
| Combination | 2 (3.6) | 3 (5.3) | 0 |
| Other | 4 (7.1) | 2 (3.5) | 1 (6.7) |
| Sick leave, n (%) | 9 (16.1) | 7 (12.3) | 3 (20.0) |
| Manage financially, n (%) | | | |
| Very well | 22 (39.3) | 34 (59.6) | 7 (46.7) |
| Quite well | 30 (53.6) | 21 (36.8) | 7 (46.7) |
| Neither well or badly | 4 (7.1) | 2 (3.5) | 1 (6.7) |
| Quite badly | 0 | 0 | 0 |
| Very badly | 0 | 0 | 0 |
| LBP/PGP last 4 weeks, n (%) | 56 (100) | 57 (100) | 15 (100) |
| Pain affects daily activities, n (%) | 53 (94.6) | 57 (100) | 15 (100) |
| Pain how often, n (%) | | | |
| Some days | 3 (5.4) | 7 (12.3) | 0 |
| Most days | 16 (28.6) | 19 (33.3) | 8 (53.3) |
| Every day | 37 (66.1) | 31 (54.4) | 7 (46.7) |
| Use of analgesics, n (%) | 6 (10.9), n=55 | 3 (5.3) | 0 |
| ODI, mean (SD), median (min; max) | 34.5 (13.19) 34.0 (8.0; 68.0) | 30.1 (11.82) 30.0 (8.0; 60.0) | 32.5 (12.22) 32.0 (12.0; 52.0) |
| PSFS NRS, mean (SD), median (min; max) | 2.8 (1.46) 2.5 (0; 7.0) | 3.2 (1.55) 3.0 (0; 7.5) | 3.9 (2.33) 3.0 (0.5; 9.0) |
| Work ability NRS, mean (SD), median (min; max) | 6.2 (2.36) 7.0 (0; 10.0), n=55 | 6.7 (2.41) 7.0 (0; 10.0), n=56 | 5.7 (3.02) 6.0 (0; 9.0) |
| Recommended PA; yes, n (%) | 13 (23.2) | 14 (24.6) | 6 (40) |
| PGQ total, mean (SD), median (min; max) | 49.3 (17.70) 50.7 (8.0; 90.7), n=55 | 47.0 (12.91) 45.6 (22.2; 82.6), n=56 | 53.2 (18.1) 60.0 (20.0; 77.3) |
| PGQ activity, mean (SD), median (min; max) | 48.1 (17.76) 50.0 (6.7; 90.0), n=55 | 46.3 (13.55) 45.3 (24.6; 82.5), n=56 | 52.1 (18.63) 58.3 (16.7; 76.7) |
| PGQ symptom, mean (SD), median (min; max) | 51.5 (17.17) 53.3 (13.3; 93.3), n=55 | 49.4 (13.27) 46.7 (13.3; 73.3) | 57.1 (15.98) 60.0 (26.67; 80.0) |

Continued

**Table 1** Continued

| Variable | Acupuncture, n=56 | TENS, n=57 | Excluded after baseline test, n=15 |
|---|---|---|---|
| Evening pain intensity NRS, mean (SD), median (min; max) | 6.8 (1.95)<br>7.0 (2.0; 10.0), n=55 | 6.6 (1.56)<br>7.0 (3.0; 9.0) | 7.4 (1.56)<br>8.0 (4.0; 9.0) |
| Concern about pain NRS, mean (SD), median (min; max) | 4.2 (2.44)<br>5.0 (0; 10.0) | 4.6 (2.34)<br>5.0 (0; 10.0) | 4.7 (2.69)<br>5.0 (0; 10.0) |
| EQ5D, mean (SD), median (min; max) | 0.55 (0.28)<br>0.69 (–0.13; 0.85) | 0.63 (0.19)<br>0.69 (–0.02; 0.89), n=55 | 0.55 (0.25)<br>0.66 (0.06; 0.8) |
| EQ VAS, mean (SD), median (min; max) | 61.1 (16.27)<br>65.0 (20.0; 90.0) | 64.4 (17.17)<br>70.0 (17.0; 97.0) | 60.1 (12.69)<br>60.0 (40.0; 80.0) |
| EPDS score, mean (SD), median (min; max) | 7.6 (4.11)<br>7.0 (1.0; 19.0), n=54 | 8.0 (4.97)<br>8.0 (0; 18.0), n=56 | 7.6 (3.92)<br>7.5 (2.0; 14.0) |
| CSQ-CAT, mean (SD), median (min; max) | 8.0 (6.65)<br>7.0 (0; 27.0) | 6.7 (4.98)<br>7.0 (0; 24.0) | 8.2 (6.41)<br>9.0 (0; 20.0) |

BMI, body mass index (kg/m$^2$); CSQ-CAT, Coping Strategies Questionnaire subscale 'Catastrophizing'; EPDS, Edinburgh Postnatal Depression Scale; EQ5D, EuroQol 5-dimension; LBP, low back pain; NRS, Numeric Rating Scale; ODI, Oswestry Disability Index; PA, physical activity ; PGP, pelvic girdle pain; PGQ, Pelvic Girdle Questionnaire ; PSFS, Patient-Specific Functional Scale; TENS, transcutaneous electrical nerve stimulation; ; VAS, Visual Analogue Scale.

## Outcomes and follow-up

Because of the present study's focus on functioning, a broad spectrum of outcome variables was included. All outcomes were measured at baseline and repeated at follow-up.

### Primary outcomes

▶ Disability, defined as affected function due to PGP, measured by ODI, range 0%–100%, where 0%=no disability.[35]
▶ Functional status, assessed by the PSFS. Two self-chosen everyday activities rated on a Numeric Rating Scale (NRS) of 0–10; 10=can perform the activity unrestrictedly or as before the onset of PGP.[36]
▶ Work ability, measured by one question from the Work Ability Index. Current work ability marked on a NRS of 0–10; 10=work ability at its best.[37]
▶ Physical activity level, number of days/week with activity for 30 min at moderate intensity and/or activity at high intensity for 20 min.[38]

### Secondary outcomes

▶ Everyday functioning affected by PGP, measured by the Pelvic Girdle Questionnaire (PGQ). A total score and scores for the subscales of 'Activity' and 'Symptoms', where 0%=no disability.[39 40]
▶ Evening PGP intensity at its worst, measured by a NRS of 0–10; 10=worst pain,[41] and by a pain drawing to illustrate the pain location.
▶ Concern about PGP, assessed using a NRS of 0–10; 10=extremely concerned.[42]
▶ Health, measured by the EuroQol 5-dimension questionnaire (EQ5D), a total score of 1 indicates full health. Participants also marked their current health status on the EuroQol vertical Visual Analogue Scale; 100=best imaginable health state.[43]

▶ Symptoms of depression, measured by the Edinburgh Postnatal Depression Scale, range 0–30. A score >13 indicates symptoms of depression that require further examination.[44 45]
▶ Catastrophising beliefs via the subscale 'Catastrophizing' (from the Coping Strategies Questionnaire, CSQ-CAT) range 0–36; 0=no catastrophising beliefs.[46 47]
▶ Questions were also asked about use of analgesics, sick leave, height/weight, education level, family income and possible side effects of the treatment.

After the intervention period the clinical examination was repeated by the independent test-leader, who was blinded to the treatment allocation, and the participants completed a follow-up questionnaire.

## Patient and public involvement

No patients or members of the public was involved in design, conduction or interpretation of the study. Patient satisfaction was measured at follow-up.

## STATISTICAL ANALYSIS

### Sample size

With a β-level of 80%, an α-level of 5% and a difference in functioning between groups of 10%, measured using the ODI, a sample size of at least 30 women/group was required. As the dropout rate for the postpartum follow-up and the planned 3.5-year long-term follow-up was calculated to be 25% and 50%, respectively, the plan was to include 60 women in each group.

### Analysis

The analysis was done according to intention-to-treat (ITT), with the use of multiple imputation with 50 imputation data sets generated followed by one analysis of each

of these 50 data sets followed by a pooled analysis of the 50 tests. The imputation of each value was performed by fully conditional specifications. For continuous variables t-test was applied due to central limit theorem and for dichotomous variables change in proportion was applied. A full analysis set (FAS) was also performed, using Fisher's nonparametric permutation test, including only those women who completed the intervention. Change from baseline to follow-up is expressed as mean (SD)/ median (min; max)/ 95% CI for mean. For dichotomous variables, the $\chi^2$ test was used. The level of statistical significance was set to $p < 0.05$. The analysis was made using IBM SPSS Statistics V.24 (IBM Corp) and SAS V.9.4 (SAS Institute).

## RESULTS

Altogether 15 women, who fulfilled the inclusion criteria, were excluded after the baseline test due to difficulties in finding time for treatment or they did not show up for randomisation (figure 1). A total of 113 women were randomised into two groups (figure 1). No relevant differences between the intervention groups were found at baseline. At baseline, women in both groups reported pain in their pelvic and/or lumbar region during the last 4 weeks, which affected their daily activities for more than 1 day/week (table 1).

The sacroiliac joint was the most common pain location (93%) (table 2).

| **Table 2** Clinical tests to verify pelvic girdle pain | | | |
|---|---|---|---|
| **Variable** | **Acupuncture** n=56 | **TENS** n=57 | **Excluded after baseline test, n=15** |
| Reported pubic pain, n (%) | | | |
| No | 20 (35.7) | 20 (35.1) | 5 (33.3) |
| Yes | 36 (64.3) | 37 (64.9) | 10 (67.7) |
| Reported sacroiliac pain, n (%) | | | |
| No | 4 (7.1) | 4 (7.0) | 1 (6.7) |
| Yes | 52 (92.9) | 53 (93.0) | 14 (93.3) |
| Localisation sacroiliac pain, n (%) | | | |
| Unilateral | 6 (11.5) | 7 (13.0) | 2 (15.4) |
| Bilateral | 46 (88.5) | 47 (87.0) | 11 (84.6 |
| | n=52 | n=54 | n=13 |
| Positive MAT-test, n (%) | | | |
| No | 29 (51.8) | 29 (51.8) | 8 (53.3) |
| Yes | 27 (48.2) | 27 (48.2) | 7 (46.7) |
| Positive 4P-test, n (%) | | | |
| No | 6 (10.7) | 2 (3.5) | 1 (6.6) |
| Yes, unilateral | 9 (16.1) | 16 (28.1) | 4 (26.7) |
| Yes, bilateral | 41 (73.2) | 39 (68.4) | 10 (66.7) |
| Positive SIJ-separation, n (%) | | | |
| No | 24 (42.9) | 24 (42.1) | 5 (33.3) |
| Yes | 32 (57.1) | 33 (57.9) | 10 (66.7) |
| Positive SIJ-compression, n (%) | | | |
| No | 42 (75.0) | 38 (66.7) | 10 (66.7) |
| Yes | 14 (25.0) | 19 (33.3) | 5 (33.3) |
| Positive sacrum ventral-test, n (%) | | | |
| No | 29 (51.8) | 31 (54.4) | 8 (53.3) |
| Yes | 27 (48.2) | 26 (45.6) | 7 (46.7) |
| Positive ASLR-test, n (%) | | | |
| No | 10 (17.8) | 8 (14) | 4 (26.7) |
| Yes, unilateral | 15 (26.8) | 19 (33.4) | 3 (20.0) |
| Yes, bilateral | 31 (55.4) | 30 (52.6) | 8 (53.3) |

ASLR, active straight leg raise; MAT, pulling a mat-test; 4P, posterior pain provocation-test; SIJ, sacroiliac joint; TENS, transcutaneous electrical nerve stimulation.

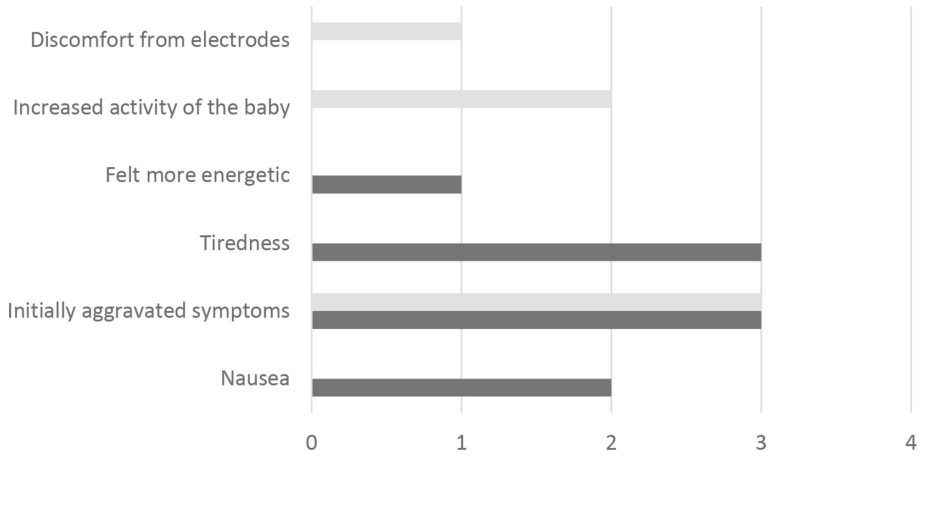

**Figure 2** Reported side effects of the interventions, acupuncture (n=9/41) and TENS (n=6/45). TENS, transcutaneous electrical nerve stimulation.

The women in both groups reported that their work ability was affected and few of them (acupuncture 23.2%; TENS 24.6%) met the general recommendations for PA. The consumption of analgesics was low in both groups (table 1).

In both groups, participants were lost to follow-up due to discontinuing the intervention (figure 1). Reasons for this was illness, vacation, need to care for younger children, difficulties to fit acupuncture sessions into work schedule or just dropping off. For acupuncture, the mean number (min;max) of treatment sessions was 9.9 (9;10) and women in the TENS group used the device for mean 26 (3;37) out of 35 days. There was no need to change the stimulation type due to insufficient effect for any of the participants in the TENS group. In the acupuncture group, 82.5% experienced positive effects of the treatment compared with 60% in the TENS group (p=0.032). At follow-up, 9 of 41 women in the acupuncture group and 6 of 45 women in the TENS group reported some side effects during the intervention (figure 2).

### Between group comparisons
No mean differences between groups were detected regarding the primary (disability, functioning, work ability, PA) or secondary outcome variables. Minor differences between the ITT and FAS were found (tables 3 and 4).

### Within group comparisons
For ODI or the level of PA, no significant change from baseline to follow-up was detected for any group. Both groups experienced a significant decrease in work ability at follow-up. The acupuncture group showed a significant improvement in functioning, measured by PSFS, mean change 0.82 (p=0.048) (tables 3 and 4).

Both groups reported significantly reduced evening pain intensity (acupuncture −0.96, p=0.049 and TENS −1.29, p=0.003) and concerns about pain (acupuncture

−1.44, p=0.0012 and TENS −1.99, p<0.001) at follow-up. (tables 3 and 4).

There were only minor differences between the ITT and FAS and this did not affect the results except for ODI and CSQ-CAT. For ODI the TENS group showed a significant increase of disability in the FAS but this was not the case in the ITT analysis. Regarding CSQ-CAT the acupuncture group, according to the FAS, reported a significant decrease in catastrophising beliefs, this was not significant in the ITT.

### DISCUSSION
The results of the present study did not show any clinically important differences between the interventions regarding the primary outcome variables.

The measurements of disability (ODI) and daily function (PGQ) showed that PGP resulted in a moderate disability and affected the women's everyday life in various ways. Disability due to PGP/LBP tends to increase over time in pregnancy,[14] but in the present study no difference at follow-up was detected. By PSFS, the functional status was assessed in an individually tailored way as the women themselves chose the measured activities. The reported activities covered, for example, housework, childcare, transportation, specific working tasks and leisure activities. There was a significant difference on PSFS at follow-up for the acupuncture group, although it may have a small clinical relevance. For PSFS, the minimal clinical important difference (MCID) varies between 0.8 (small change) and 3.0 in different studies and for different conditions.[48] At present, there are no studies that have evaluated PSFS in pregnancy. Both groups experienced a significant decrease of work ability (tables 3 and 4). As previously shown, work ability is affected by a variety of factors in addition to PGP as pregnancy proceeds such as work conditions, general health, bodily changes and

**Table 3** Change within groups and difference in change between the groups acupuncture and TENS (intention-to-treat analysis, multiple imputation of 50 imputed data sets)

| Outcomes | Acupuncture | | TENS | | Difference between groups | |
|---|---|---|---|---|---|---|
| | Mean change (95% CI) | P value | Mean change (95% CI) | P value | Mean difference (95% CI) | P value |
| ODI | 0.44 (−3.74 to 4.61) | 0.84 | 3.53 (−0.79 to 7.85) | 0.11 | −3.09 (−9.36 to 3.17) | 0.33 |
| PSFS | 0.82 (0.01 to 1.63) | 0.048 | −0.16 (−0.79 to 0.47) | 0.62 | 0.98 (−0.03 to 1.99) | 0.058 |
| Work ability NRS | −1.39 (−2.54 to −0.24) | 0.018 | −1.27 (−1.92 to −0.62) | 0.0001 | −0.12 (−1.44 to 1.19) | 0.86 |
| Recommended PA | −0.03 (−0.20 to 0.14) | 0.72 | −0.03 (−0.20 to 0.13) | 0.68 | −0.01 (−0.13 to 0.11) | 0.87 |
| PGQ total | −0.57 (−6.43 to 5.29) | 0.85 | 3.08 (−2.28 to 8.43) | 0.26 | −3.65 (−11.46 to 4.16) | 0.36 |
| PGQ activity | 0.37 (−5.69 to 6.44) | 0.90 | 3.54 (−1.94 to 9.03) | 0.20 | −3.17 (−11.14 to 4.80) | 0.43 |
| PGQ symptoms | −1.34 (-8.33 to 5.65) | 0.71 | 1.73 (-4.12 to 7.58) | 0.56 | −3.07 (−12.17 to 6.04) | 0.51 |
| Evening pain NRS | −0.96 (−1.91 to −0.01) | 0.049 | −1.29 (-2.13 to −0.44) | 0.003 | 0.33 (−0.99 to 1.64) | 0.62 |
| Concern about pain NRS | −1.44 (−2.31 to −0.57) | 0.0012 | −1.99 (−2.81 to −1.17) | <0.0001 | 0.55 (−0.64 to 1.73) | 0.36 |
| EQ5D | 0.04 (−0.07 to 0.14) | 0.51 | −0.10 (−0.21 to 0.00) | 0.059 | 0.14 (−0.01 to 0.28) | 0.061 |
| EQ5D vertical VAS | −1.18 (−7.72 to 5.37) | 0.72 | −1.66 (−8.79 to 5.48) | 0.65 | 0.48 (−9.29 to 10.25) | 0.92 |
| EPDS | −0.40 (−2.07 to 1.28) | 0.64 | −1.25 (−2.77 to 0.27) | 0.11 | 0.85 (−1.36 to 3.07) | 0.45 |
| CSQ-CAT | −2.50 (−5.43 to 0.43) | 0.095 | −1.21 (−2.68 to 0.25) | 0.10 | −1.29 (−4.55 to 1.98) | 0.44 |

CSQ-CAT, Coping Strategies Questionnaire subscale 'Catastrophizing'; EPDS, Edinburgh Postnatal Depression Scale; EQ5D, EuroQol 5-dimension; NRS, Numeric Rating Scale; ODI, Oswestry Disability Index; PA, physical activity; PGQ, Pelvic Girdle Questionnaire; PSFS, Patient-Specific Functional Scale; TENS, transcutaneous electrical nerve stimulation; VAS, Visual Analogue Scale.

tiredness.[49] This result indicates that pregnancy itself has an impact on women's work ability.[14]

At follow-up, the level of PA was maintained for both groups (tables 3 and 4). This differs from previous studies that showed that the level of activity decreases as pregnancy proceeds.[11] [50–53] The interventions did reduce evening pain and concern about pain and this reduction may help women to maintain their PA level. The amount of evening pain relief in the present study could be considered small; mean change −0.96 for acupuncture and −1.29 for TENS (tables 3 and 4) but a MCID of −1.3 is reported as meaningful for pregnant women with PGP.[54] No clinically important difference could be detected between or within groups regarding health. Previous studies show that pregnant women, especially women with PGP and/or LBP, experience lower health compared with non-pregnant women[55] and also that quality of life decreases as pregnancy proceeds.[56] The results of the present study may indicate that a decrease in pain intensity and concern about pain

could result in maintenance of overall functioning in daily activities, PA and, consequently, better health.

The women experienced positive effect of the treatments, acupuncture 82.5% and TENS 60%. This difference could be explained by that the participants in the acupuncture group had more contact with the physiotherapist. The interaction between practitioner and patient may affect the outcome of the intervention[57] and we cannot clearly say that the two interventions alone caused the positive effects experienced by the participants. In this study, both interventions gave significant pain relief but no significant difference between the groups was detected. The increase of functioning (PSFS) at follow-up for the acupuncture group could, even though it may be a small change,[48] indicate that the women experienced a positive impact on everyday activities that were important for them which in turn led to a positive experience of the intervention (tables 3 and 4). A positive effect on activity level could be as relevant for the individual woman as

**Table 4** Change within groups and difference in change between the groups acupuncture and TENS (full analysis set, all available data)

| Outcome | Acupuncture (n=41) Baseline Mean (SD) Median (min; max) | Follow-up Mean (SD) Median (min; max) | Mean change (SD) Median (min; max) (95% CI for mean)* n= | P value | TENS (n=45) Baseline Mean (SD) Median (min; max) | Follow-up Mean (SD) Median (min; max) | Mean change (SD) Median (min; max) (95% CI for mean)* n= | P value | Difference between groups Mean difference (95% CI)‡ | P value |
|---|---|---|---|---|---|---|---|---|---|---|
| ODI | 32.59 (11.40) 34.00 (8.0; 60.0) | 33.07 (13.41) 34.0 (10.0; 66.0) | 0.49 (11.89) 2 (−22; 26) (−3.20 to 4.44) n=41 | 0.80 | 29.07 (10.86) 28.0 (8.0; 60.0) | 32.93 (15.24) 32.0 (2.0; 64.0) | 3.87 (12.36) 4 (−28; 34) (0.18 to 7.68) n=45 | 0.043 | −3.38 (−8.53 to 1.88) | 0.21 |
| PSFS | 2.74 (1.54) 2.5 (0; 7.0) | 3.63 (2.34) 3.5 (0; 10.0) | 0.89 (2.45) 0 (−3; 6.5) (0.11 to 1.67) n=41 | 0.025 | 3.16 (1.45) 3.0 (0.5; 7.5) | 3.14 (1.75) 3.0 (0; 8.0) | −0.017 (1.85) 0 (−4.5; 4.5) (−0.57 to 0.54) n=45 | 0.95 | 0.91 (0.00 to 1.85) | 0.053 |
| Work ability NRS | 6.45 (2.39) 7.0 (0; 10.0) | 5.34 (3.11) 6.0 (0; 10.0) | −1.10 (3.13) −1 (−8; 6) (−2.12 to −0.11) n=40 | 0.036 | 7.0 (2.28) 7.0 (0; 10.0) | 5.67 (2.54) 6.0 (0; 10) | −1.33 (1.92) −1 (−6; 2) (−1.91 to −0.76) n=45 | <0.0001 | 0.23 (−0.88 to 1.31) | 0.70 |
| Recommended PA; n (%) | 9 (22.0) | 8 (19.5) | −0.02 (0.42) 0 (−1; 1) (−0.15 to 0.11) n=41 | 1.00 | 10 (22.2) | 9 (20.0) | −0.02 (0.40) 0 (−1; 1) (−0.14 to 0.10) n=45 | 1.00 | −0.002 (−0.17 to 0.17) | 1.00 |
| PGQ total | 46.68 (16.30) 48.64 (8.0; 89.72) | 47.30 (16.32) 51.02 (12.0; 80.0) | 0.82 (16.26) 3 (−34.67; 32) (−4.51 to 6.18) n=39 | 0.76 | 45.25 (11.76) 42.16 (22.22; 69.33) | 48.30 (16.74) 47.82 (12.50; 81.94) | 3.84 (14.49) 2.05 (−38.17; 34.67) (−0.48 to 8.17) n=44 | 0.086 | −3.03 (−9.74 to 3.63) | 0.37 |
| PGQ activity | 44.93 (15.72) 48.33 (6.67; 75.0) | 47.02 (17.10) 49.12 (11.67; 78.94) | 2.37 (15.65) 3.33 (−31.76; 31.67) (−2.77 to 7.44) n=39 | 0.35 | 44.56 (12.16) 42.72 (24.56; 70.0) | 48.00 (17.05) 46.67 (10.52; 82.45) | 4.24 (14.81) 3.42 (−37.81; 33.33) (−0.21 to 8.66) n=44 | 0.064 | −1.87 (−8.50 to 4.76) | 0.58 |
| PGQ symptoms | 49.83 (16.40) 53.33 (13.33; 86.67) | 48.42 (18.10) 48.33 (13.33; 86.67) | −1.45 (20.67) 0 (−53.33; 33.33) (−8.24 to 5.21) n=39 | 0.66 | 48.00 (13.38) 46.67 (13.33; 73.33) | 49.48 (18.12) 46.67 (0; 86.67) | 1.48 (17.69) 6.67 (−40; 40) (−3.86 to 6.67) n=45 | 0.58 | −2.93 (−11.33 to 5.56) | 0.50 |
| Evening pain NRS | 6.82 (1.8) 7.0 (2.0; 9.5) | 5.58 (2.26) 6.0 (1.0; 9.0) | 1.26 (2.64) 1 (−7; 4) (−2.12 to −0.40) n=39 | 0.0052 | 6.60 (1.5) 7.0 (3.0; 9.0) | 5.33 (2.36) 6.0 (0; 9.0) | −1.27 (2.27) −1 (−9; 3) (−1.94 to −0.60) n=45 | 0.0002 | 0.01 (−1.06 to 1.07) | 1.00 |
| Concern about pain NRS | 3.95 (2.24) 5.0 (0; 8.0) | 2.43 (2.15) 2.0 (0; 7.0) | −1.45 (2.43) −2 (−7; 6) (−2.25 to −0.68) n=40 | 0.0007 | 4.62 (2.41) 5.0 (0; 10.0) | 2.62 (2.20) 2.0 (0; 8.0) | −2.00 (2.52) −2 (−8; 2) (−2.77 to −1.25) n=45 | <0.0001 | 0.55 (−0.52 to 1.63) | 0.32 |
| EQ5D | 0.58 (0.25) 0.69 (−0.13; 0.85) | 0.62 (0.23) 0.69 (−0.07; 0.88) | 0.03 (0.32) 0 (−0.76; 0.75) (−0.08 to 0.13) n=39 | 0.61 | 0.65 (0.18) 0.69 (−0.02; 0.89) | 0.55 (0.29) 0.66 (−0.13; 1.0) | −0.10 (0.31) −0.04 (−0.91; 0.61) (−0.20 to −0.01) n=44 | 0.036 | 0.13 (−0.01 to 0.27) | 0.073 |
| EQ5D vertical VAS | 62.24 (16.39) 65.0 (30.0; 90.0) | 63.17 (16.38) 65.0 (25.0; 90.0) | 0.93 (20.27) 0 (−47; 50) (−5.60 to 7.28) n=41 | 0.78 | 66.20 (15.16) 70.0 (27.0; 95.0) | 63.78 (19.51) 70.0 (19.0; 90.0) | −2.42 (19.75) −1 (−55; 47) (−8.25 to 3.45) n=45 | 0.42 | 3.35 (−5.31 to 11.75) | 0.45 |
| EPDS | 6.83 (3.99) 6.0 (1.0; 15.0) | 6.15 (4.73) 5.0 (0; 15.0) | −0.85 (4.27) −1 (−10; 11) (−2.25 to 0.52) n=39 | 0.24 | 7.38 (4.91) 6.0 (0; 18.0) | 6.27 (4.93) 5.0 (0; 21.0) | −1.11 (4.46) 0 (−12; 9) (−2.43 to 0.24) n=45 | 0.11 | 0.27 (−1.65 to 2.17) | 0.79 |

Continued

**Table 4** Continued

| Outcome | Acupuncture (n=41) | | | | TENS (n=45) | | | | Difference between groups | |
|---|---|---|---|---|---|---|---|---|---|---|
| | Baseline Mean (SD) Median (min; max) | Follow-up Mean (SD) Median (min; max) | Mean change (SD) Median (min; max) (95% CI for mean)* n= | P value | Baseline Mean (SD) Median (min; max) | Follow-up Mean (SD) Median (min; max) | Mean change (SD) Median (min; max) (95% CI for mean)* n= | P value | Mean difference † (95% CI)‡ | P value |
| CSQ-CAT | 7.15 (5.98) 6.0 (0; 27) | 4.49 (5.44) 2.0 (0; 18.0) | −2.60 (6.00) −2.5 (−15; 12) (−4.50 to −0.70) n=40 | 0.010 | 6.07 (4.83) 6.0 (0; 24.0) | 5.11 (5.84) 4.0 (0; 28.0) | −1.09 (4.50) −1 (−10; 17) (−2.44 to 0.27) n=44 | 0.12 | −1.51 (−3.82 to 0.78) | 0.20 |

*For comparison within groups the Fisher's non-parametric permutation test for matched pairs was used.
†For comparison between groups the Fisher's non-parametric permutation test was used for continuous variables.
‡The CI for then mean difference between groups is based on Fisher's non-parametric permutation test.
CSQ-CAT, Coping Strategies Questionnaire subscale 'Catastrophizing'; EPDS, Edinburgh Postnatal Depression Scale; EQ5D, EuroQol 5-dimension; NRS, Numeric Rating Scale; ODI, Oswestry Disability Index; PA, physical activity; PGQ, Pelvic Girdle Questionnaire; PSFS, Patient-Specific Functional Scale; TENS, transcutaneous electrical nerve stimulation; ; VAS, Visual Analogue Scale.

pain relief, which could be important to consider when treatment choices are made.

Women with PGP experience a loss of independence as the pain makes them rely on other people for help, for example, with childcare and household tasks.[24] Treatment with TENS gives a woman control over her pain relief; she can use it where or when she wants. By contrast, for acupuncture, she must make appointments and go to a clinic for treatment. Some women may need more support and may benefit from regular visits to a physiotherapy clinic whereas others feel confident to perform self-treatment after receiving instructions and information about their reports. The present study indicates that treatment of PGP during pregnancy can make women feel less concerned about their pain. This may lead to less distress and better ability to cope with and manage the pain. Women experiencing higher levels of anxiety/depressive symptoms reported higher levels of disability compared with women with lower levels[19] and anxiety predicted lower levels of PA during pregnancy.[58] These are factors important to identify and manage to prevent the development of longstanding PGP that occur in 10% of women who experienced PGP during pregnancy.[8 9]

### Strengths and limitations

Strengths of the present study are that the interventions were evaluated in both the second and the third trimester and that PGP was clinically verified in all participants. Another strength is the compliance among those who completed the interventions. In the acupuncture group, 38 out of 41 women (93 %) had the assigned 10 sessions (mean 9.9). One participant experienced pain relief after six sessions and the intervention was not continued due to ethical considerations, two participants missed a session each due to business travel and vacation. The women in the TENS group used the treatment at home for mean 26 out of 35 possible days during the treatment period. Only a few minor side effects were reported (figure 2), mostly at the beginning of the intervention period and none of them made the women discontinue treatment. Both TENS and acupuncture are according to previous studies safe to use during pregnancy.[25 27]

It is possible that the design of the present study, where the women had to travel to a specific physiotherapy clinic for testing, acupuncture and follow-up, made women with more severe reports decline to participate.[59] Another limitation of the present study is the self-reported use of TENS. Since the treatment was not monitored, it is possible that the women applying TENS treatment at home did not follow the instructions even though they followed protocol by registering the sessions. Previous research has reported that there is a risk for developing tolerance against the analgesic effect of TENS after a few days and a solution for this could be to change electrode placement and/or mode of stimulation during the treatment period.[60 61] In the present study, the mode of stimulation or the electrode placement could be changed by the treating physiotherapist after 1 week if the woman

had insufficient analgesic effect, but this was not needed for any of the participants. Nevertheless, the participants experienced positive effects of the treatment.

## CONCLUSION

No differences regarding the effect of acupuncture or TENS as treatment for PGP in pregnancy were detected in the present study. However, both interventions resulted in maintained PA level as well as reduced evening pain and concern about pain. The acupuncture group showed a change in functioning at follow-up as well as higher satisfaction with treatment. This study suggests that, based on individual women's preferences and needs, these interventions could be chosen as non-pharmacological alternatives for pain relief in order to maintain functioning and PA level and consequently, maintain health as pregnancy proceeds.

**Acknowledgements** The authors thank the participating pregnant women, the midwives for their help in recruiting the participants, the physiotherapists that provided the treatments and Nils-Gunnar Pehrsson for statistical advice.

**Contributors** Concept/idea/research design: AG, MFO, TJ. Writing: ASE, AG, MFO. Data collection: ASE, TJ. Data analysis: ASE, AG, MFO. Project management: AG, MFO, ASE. Consultation (including review of manuscript before submitting): AG, MFO, TJ.

**Funding** This study was financed by Region Västra Götaland, Sweden No. VGFOUREG-310521. The sponsor of the study did not have any influence on study design, collection and analysis/interpretation of data or writing the manuscript.

**Competing interests** None declared.

**Patient and public involvement** Patients and/or the public were not involved in the design, or conduct, or reporting, or dissemination plans of this research.

**Patient consent for publication** Not applicable.

**Ethics approval** The study was approved by the regional ethical review board in Gothenburg, Sweden (No. 308–13).

**Provenance and peer review** Not commissioned; externally peer reviewed.

**Data availability statement** Data are available upon reasonable request. The data on which the analyses presented in this manuscript are based, are available upon reasonable request from the corresponding author.

**ORCID iD**
Annika Svahn Ekdahl http://orcid.org/0000-0001-5665-0553

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
