## [Reviewer comments · BMJ Open]

ARTICLE DETAILS

TITLE (PROVISIONAL)	Maintenance of physical activity level, functioning and health after non-pharmacological treatment of pelvic girdle pain with either transcutaneous electric nerve stimulation or acupuncture. A randomized controlled trial
AUTHORS	Svahn Ek Dahl, Annika; Olsen, Monika Fagevik; Jendman, Tove; Gutke, Annelie

VERSION 1 – REVIEW

REVIEWER	Comachio, Josielli The University of Sydney Faculty of Medicine and Health
REVIEW RETURNED	11-Jan-2021

GENERAL COMMENTS	In general, the article is good. However, I encourage the authors re-write the paper and add more information that is extremely necessary in a RCT. I think it is a very interesting paper and it is important with studies like this that investigate pelvic girdle pain with either transcutaneous electric nerve stimulation or acupuncture. The first part of the paper is easy to read and the introduction explains the necessary terms and builds a good argumentation for the intervention and the aim of the study. Nevertheless, while reading, there is no attention to process variables, e.g., such as number of sessions attended, position of patient during session, localisation of meridians, types of needles, localisation of TENS, adherence with TENS and, if homework advises (if existed) which can impact treatment outcomes. Furthermore, please following the instruction based (STRICTA): extending the CONSORT statement. The STRICTA checklist consists of six items split into 17 sub-items. These items include details of the rationale for acupuncture, details of needling, treatment ... it should be submitted together. In addition, the outcomes are missing the definition of the construct that is being measured. Consequentially, it is not clear what, for example, form of disability is being referred to. I infer from the outcome measures that this is pain girdle. I encourage the authors to re-phrase this as Pelvic Girdle Questionnaire (and to define each outcome using: domain, construct, measure(s), time point(s). Disability might thus be written as 'Function, defined as low back-girdle function, measured using the ODI,, at xxxxx (time point). Cordially,
--

REVIEWER	Barlas, Panos Metro North Hospital and Health Service, Jamieson Trauma Institute
-----------------	---

REVIEW RETURNED	19-Feb-2021
-------------

GENERAL COMMENTS	The trial details two simple interventions for the management of a persistent and bothersome problem for women of childbearing age. It is described clearly and its methods are correct. It would have been useful to have collected and presented data for a longer follow-up period than the one reported which was only 5 weeks. One question that has not been addressed completely within the text is the dosage of TENS treatment. Whilst there is mention of the frequency (80Hz) there is no discussion on the instructions for the intensity of stimulation which is a significant factor for a maximum analgesic effect. Perhaps a sentence or two discussing the intensity of TENS would be appropriate (pg. 8 lines 50-53). Additionally (and a minor point) the authors may wish to revise 'sensibility' for 'sensation' (pg. 7 Line 58). Moreover the authors seem to not have taken into account the possibility of developing tolerance to TENS (which may also explain why the reported analgesic effect was lower than that of acupuncture). See: https://pubmed.ncbi.nlm.nih.gov/21144659/. Other than the points above, which I suggest the authors do address, I think that this study conveys a significant message and I sincerely hope its recommendations are heard.
--

REVIEWER	Betts, Debra New Zealand School of Acupuncture and Traditional Chinese Medicine
REVIEW RETURNED	14-Apr-2021

GENERAL COMMENTS	This is an interesting and valuable paper that contributes to the body of knowledge for treatment options for pelvic pain in pregnancy. However, the following are concerns that require addressing. Abstract Results Section – Amendment to report finding that statistical significance was found for (Patient Specific Functional Scale) in the acupuncture group as this is a primary outcome measure. Required justification why the finding that the acupuncture group, 82.5% experienced positive effects of the treatment compared to 60% in the TENS group (p=0.032) is not reported in abstract. Although as later clarified in the paper this may have been due to other variables such as increased contact with the treatment provider it a finding that is relevant for the abstract and for further research to explore why the lower satisfaction in the Tens group. Within Paper P 8 Line 38 “It was as decided to shorten the treatment if any woman experienced enough pain relief in fewer than ten sessions.” Unclear – does this refer to a shorter than 30-minute treatment? And if so, what did this treatment involve? Or does it refer to shortening the number of treatments received? If so, later reporting (lines below), require clarification – did the three women (7%) have reduced treatment due to beneficial effects requiring no further treatment? P 3 Line 15 P 20 lines 73-74 In the acupuncture group, 38 out of 41 women (93 %) had the assigned ten sessions. Intervention for TENS
--

	It is unclear if other practitioners would be able to repeat intervention from the description “placing the electrodes in the area of pain and using high frequency stimulation (80 Hz).” Many women have radiating pain into groin, directly over the pubic synthesis and into their thigh area. Were electrodes placed on suggested areas in oral consultations and written instructions? Including information given to women as supplementary data would be useful. Page 12 Results section According to Figure 1. After commencing treatment 11 (20%) in the acupuncture group and 6 (11%) in the TENS group that discontinued the intervention. Was any data collected that accounts for the participants reasons for this? P 19 lines 23- 24 “The results of the present study did not show any clinically important differences between the interventions regarding the primary outcome variables.” This requires amendment as PSFS, was a primary outcome. Although not an objective functional measurement it is comparable to the three other primary outcome measures where women were self-assessing work ability and physical level ability. The finding that there were statistically significant differences for women in the acupuncture group can be seen as very clinically relevant for women. With self -selected activities such as housework, childcare, transportation, specific working tasks and leisure activities are an important consideration for women when choosing treatment options. Page 19 Lines 54 “In this study, both interventions gave significant pain relief but no significant difference between the groups was detected (Table 3, 4), which indicates that the treatment can be chosen based on the individual woman’s needs and preferences.” As above, required amendment as incorrect. Statistically significant differences were found for PSFS relating to self -selected activities. In the context of the discussion below, it may have been relevant that the positive effect of the treatment was also from the women’s self-reporting of improvement in activities such as housework, childcare, transportation, specific working tasks and leisure activities and may be just as relevant as pain relief for many women. “The women experienced positive effect of the treatments, acupuncture 82.5 % and TENS 60%. This difference could be explained by that the participants in the acupuncture group had more contact with the physiotherapist. The interaction between practitioner and patient may affect the outcome of the intervention and we cannot clearly say that the two interventions alone caused the positive effects experienced by the participants. In this study, both interventions gave significant pain relief but no significant difference between the groups was detected (Table 3, 4), which indicates that the treatment can be chosen based on the individual woman’s needs and preferences.” Page 21 Line 88 -89 “No differences regarding the effect of acupuncture or TENS as treatment for PGP in pregnancy were detected in the present study. “
--	---

	This statement ignores the findings of Statistically significant differences were found for PSFS relating to self -selected activities and the reporting that 'women experienced positive effect of the treatments acupuncture compared top TENS 82.5 % Discussion Section Within the discussion section it would be useful to clarify the difference between the acupuncture and TENS treatment – As it currently reads the acupuncture points were taken from a protocol using local and distal acupuncture points – while the TENS protocol asked women to select local areas of pain. This would clarify the any misconception that the acupuncture protocol and TENS protocol were using similar points.
--	--

VERSION 1 – AUTHOR RESPONSE

REVIEWERS' COMMENTS

Authors' reply: Thank you for all work to review our manuscript and all valuable comments that have improved the manuscript. A point-by-point response is stated below, and changes made in the manuscript are highlighted in yellow.

Reviewer: 1

Ms. Josielli Comachio, University of Sao Paulo Comments to the Author:

In general, the article is good. However, I encourage the authors re-write the paper and add more information that is extremely necessary in a RCT. I think it is a very interesting paper and it is important with studies like this that investigate pelvic girdle pain with either transcutaneous electric nerve stimulation or acupuncture.

The first part of the paper is easy to read and the introduction explains the necessary terms and builds a good argumentation for the intervention and the aim of the study. Nevertheless, while reading, there is no attention to process variables, e.g., such as number of sessions attended, position of patient during session, localisation of meridians, types of needles, localisation of TENS, adherence with TENS and, if homework advises (if existed) which can impact treatment outcomes.

Reply: Thank you for this comment, we agree that this section could be more specific, and revised this section in the manuscript. (p. 8-9 lines 159-184)

Furthermore, please following the instruction based (STRICTA): extending the CONSORT statement. The STRICTA checklist consists of six items split into 17 sub-items. These items include details of the rationale for acupuncture, details of needling, treatment ... it should be submitted together.

Reply: We have submitted the STRICTA checklist together with the CONSORT checklist including the Non-pharmacological Trials Extension along with the revised manuscript, thank you for pointing this out.

In addition, the outcomes are missing the definition of the construct that is being measured. Consequentially, it is not clear what, for example, form of disability is being referred to. I infer from the outcome measures that this is pain girdle. I encourage the authors to re-phrase this as Pelvic Girdle Questionnaire (and to define each outcome using: domain, construct, measure(s), time point(s)).

Disability might thus be written as 'Function, defined as low back-girdle function, measured using the ODI,, at xxxxx (time point).

Reply: Thank you for this suggestion, we have revised the section in the manuscript accordingly. (p. 9 lines 189-191 and p. 10 lines 201-206)

—

Reviewer: 2

Dr. Panos Barlas, Metro North Hospital and Health Service Comments to the Author:

The trial details two simple interventions for the management of a persistent and bothersome problem for women of childbearing age.

It is described clearly and its methods are correct. It would have been useful to have collected and presented data for a longer follow-up period than the one reported which was only 5 weeks.

Reply: Thank you for this highly relevant comment. In this study, our aim was to investigate if only a short intervention period for PGP may affect pregnant women's physical activity level, functioning and health. In the clinical setting, 5 weeks may be a common period for physiotherapy treatment, so we wanted to evaluate the interventions for just a short period. We have data from a follow up 4 months post-partum and we are also collecting data for a 3-year follow up. This will be presented in a future paper.

One question that has not been addressed completely within the text is the dosage of TENS treatment. Whilst there is mention of the frequency (80Hz) there is no discussion on the instructions for the intensity of stimulation which is a significant factor for a maximum analgesic effect. Perhaps a sentence or two discussing the intensity of TENS would be appropriate (pg. 8 lines 50-53).

Reply: We agree that this could be expressed in more detail and more information were added in the manuscript. (pg. 9 lines 173-184)

Additionally (and a minor point) the authors may wish to revise 'sensibility' for 'sensation' (pg. 7 Line 58).

Reply: We thank you for clarifying this, it has been changed in the manuscript. (pg. 7 line 146)

Moreover the authors seem to not have taken into account the possibility of developing tolerance to TENS (which may also explain why the reported analgesic effect was lower than that of acupuncture). See: <https://pubmed.ncbi.nlm.nih.gov/21144659/>.

Reply: Thank you for pointing out this interesting aspect of tolerance to TENS and the link to a relevant and useful article. The development of tolerance is something that probably could be a part of discussing differences in reported analgesic effect and we have added this aspect to the Discussion section of the manuscript and included the suggested reference and this newly published article <https://pubmed.ncbi.nlm.nih.gov/33919821/> (pg 21 lines 382-387)

Other than the points above, which I suggest the authors do address, I think that this study conveys a significant message and I sincerely hope its recommendations are heard.

–

Reviewer: 3

Dr. Debra Betts, New Zealand School of Acupuncture and Traditional Chinese Medicine Comments to the Author:

This is an interesting and valuable paper that contributes to the body of knowledge for treatment options for pelvic pain in pregnancy.

However, the following are concerns that require addressing.

Abstract

Results Section – Amendment to report finding that statistical significance was found for (Patient Specific Functional Scale) in the acupuncture group as this is a primary outcome measure.

Reply: Thank you for this relevant comment. In the abstract we reported that no mean differences were found between groups for any of the variables. For PSFS, there was a significant difference for the acupuncture group, the MCID for PSFS varies between 0.8 (small change) and 3.0 in different studies and for different conditions (Horn, K. K. et al.

2012 <https://pubmed.ncbi.nlm.nih.gov/22031594/>). We agree that there is a statistically significant difference in the acupuncture group regarding PSFS, but this difference may not be clinically important. However, it is a statistical significance within the acupuncture group, and we have revised the manuscript and clarified how this result was presented in the abstract. (p.3 lines 53-55)

Required justification why the finding that the acupuncture group, 82.5% experienced positive effects of the treatment compared to 60% in the TENS group ($p=0.032$) is not reported in abstract. Although as later clarified in the paper this may have been due to other variables such as increased contact with the treatment provider it a finding that is relevant for the abstract and for further research to explore why the lower satisfaction in the Tens group.

Reply: Thank you for pointing this out, we addressed this further in the Discussion section as suggested and reported the difference also in the “Strengths and Limitations” section after the abstract (p. 4 lines 69-70)

Within Paper

P 8

Line 38

“It was as decided to shorten the treatment if any woman experienced enough pain relief in fewer than ten sessions.”

Unclear – does this refer to a shorter than 30-minute treatment? And if so, what did this treatment involve? Or does it refer to shortening the number of treatments received? If so, later reporting (lines below), require clarification – did the three women (7%) have reduced treatment due to beneficial effects requiring no further treatment?

P 3 Line 15

P 20 lines 73-74 In the acupuncture group, 38 out of 41 women (93 %) had the assigned ten sessions.

Reply: Our decision was that if any participant experienced sufficient pain relief after fewer than 10 acupuncture sessions, it was not ethical to continue the intervention. This was the case for one woman, who experienced that she was pain-free after 6 treatment sessions. The other two women missed one session each due to business travel or vacation. We have clarified this in the manuscript (p. 8 lines 163-165 and p. 20 lines 369-371)

Intervention for TENS

It is unclear if other practitioners would be able to repeat intervention from the description “placing the electrodes in the area of pain and using high frequency stimulation (80 Hz).” Many women have radiating pain into groin, directly over the pubic synthesis and into their thigh area. Were electrodes placed on suggested areas in oral consultations and written instructions? Including information given to women as supplementary data would be useful.

Reply: Thank you for this suggestion for clarification. We agree that this section could be more specific, and we have revised the Intervention section in the manuscript. (p.9 lines 173-184) In this study the aim was to investigate TENS and acupuncture in the clinical everyday setting anthe treating physiotherapist decided, according to the woman’s symptoms, which electrode placement that should be used. This means that the electrode placement could differ between individuals within the frames of the protocol.

Page 12 Results section

According to Figure 1. After commencing treatment 11 (20%) in the acupuncture group and 6 (11%) in the TENS group that discontinued the intervention. Was any data collected that accounts for the participants reasons for this?

Reply: Thank you for this relevant question. Yes, there is data on the reasons for discontinuing the interventions and this was due to for example illness, going on vacation, the need to care for younger children, difficulties to fit acupuncture sessions into work schedules or just dropping off. We have added this information in the manuscript. (p. 14 lines 266-268)

P 19 lines 23- 24

“The results of the present study did not show any clinically important differences between the interventions regarding the primary outcome variables.”

This requires amendment as PSFS, was a primary outcome. Although not an objective functional measurement it is comparable to the three other primary outcome measures where women were self-assessing work ability and physical level ability. The finding that there were statistically significant differences for women in the acupuncture group can be seen as very clinically relevant for women. With self -selected activities such as housework, childcare, transportation, specific working tasks and leisure activities are an important consideration for women when choosing treatment options.

Reply: Thank you, this comment is relevant and as we replied regarding your comment about reporting PSFS results in the abstract, we have revised the way the data for PSFS was reported to avoid misconceptions. There were no statistically differences regarding any of the primary variables between the groups but for PSFS there was a slight improvement within group at follow up. (p. 18-19 lines 321-324)

Page 19 Lines 54

“In this study, both interventions gave significant pain relief but no significant difference between the groups was detected (Table 3, 4), which indicates that the treatment can be chosen based on the individual woman’s needs and preferences.”

As above, required amendment as incorrect. Statistically significant differences were found for PSFS relating to self -selected activities. In the context of the discussion below, it may have been relevant

that the positive effect of the treatment was also from the women’s self-reporting of improvement in activities such as housework, childcare, transportation, specific working tasks and leisure activities and may be just as relevant as pain relief for many women.

Reply: This comment is important and in line with the previous one, we agree that the results for PSFS could be reported in a more specific way and we revised this section in the manuscript. (p. 19-20 lines 347-351)

“The women experienced positive effect of the treatments, acupuncture 82.5 % and TENS 60%. This difference could be explained by that the participants in the acupuncture group had more contact with the physiotherapist. The interaction between practitioner and patient may affect the outcome of the intervention and we cannot clearly say that the two interventions alone caused the positive effects experienced by the participants. In this study, both interventions gave significant pain relief but no significant difference between the groups was detected (Table 3, 4), which indicates that the treatment can be chosen based on the individual woman’s needs and preferences.”

Page 21 Line 88 -89

“No differences regarding the effect of acupuncture or TENS as treatment for PGP in pregnancy were detected in the present study. “

This statement ignores the findings of Statistically significant differences were found for PSFS relating to self -selected activities and the reporting that ‘women experienced positive effect of the treatments acupuncture compared top TENS 82.5 %

Reply: No differences between groups were detected but we agree that the statistical findings for PSFS regarding the within group results, although they may have a small clinical relevance, as well as the experienced positive effect of acupuncture may need to be presented more clearly, this is revised in the manuscript. (p. 21 lines 392-394)

Discussion Section

Within the discussion section it would be useful to clarify the difference between the acupuncture and TENS treatment – As it currently reads the acupuncture points were taken from a protocol using local and distal acupuncture points – while the TENS protocol asked women to select local areas of pain. This would clarify the any misconception that the acupuncture protocol and TENS protocol were using similar points.

Reply: Thank you for this useful suggestion. We have revised the “Randomization and Interventions” section and added information that we hope clarified this comment. (p. 8-9 lines 162-184)

VERSION 2 – REVIEW

REVIEWER	Comachio, Josielli The University of Sydney Faculty of Medicine and Health
REVIEW RETURNED	01-Jun-2021

GENERAL COMMENTS	Dear author, Thank you for the opportunity to review this interesting manuscript. I have a few comments that I list below: I could not find the hypotheses of the study on pages 3-4 as listed. Can you please add a hypothesis? Can you explain in more detail the location of electrodes in TENS group? I could not find this information on methods. Page 19. #303, The difference between groups chosen for the sample size calculation (1 point) seems quite small, especially for
---

	the PSFS. Is that clinically relevant for this population? Where were the SDs estimated from? Can you please provide references? Page 30. Figure 2. Despite the table showed the effects sizes of both interventions. It is clear the majority of effects sizes is in the group Acupuncture. Do you have any reasons for that? The acupoints selected possible can interfere into the results expectations? In addition, any recommendations were delivered due to the effects sides in pregnancy?
--	--

REVIEWER	Betts, Debra New Zealand School of Acupuncture and Traditional Chinese Medicine
REVIEW RETURNED	30-May-2021

GENERAL COMMENTS	Thank you for submitting this review and clarifying previous reviewers comments
---

VERSION 2 – AUTHOR RESPONSE

REVIEWERS' COMMENTS

Authors' reply: Thank you for taking your time to review our manuscript and for valuable comments to improve the manuscript. A point-by-point response is stated below, and changes made in the manuscript are highlighted in yellow.

Reviewer: 3

Dr. Debra Betts, New Zealand School of Acupuncture and Traditional Chinese Medicine Comments to the Author:

Thank you for submitting this review and clarifying previous reviewers comments

Reviewer: 1

Ms. Josielli Comachio, The University of Sydney Faculty of Medicine and Health Comments to the Author:

Dear author,

Thank you for the opportunity to review this interesting manuscript. I have a few comments that I list below:

I could not find the hypotheses of the study on pages 3-4 as listed. Can you please add a hypothesis?

Reply: Our study has an explorative design because when this study was planned, there was only

one previous study that had investigated TENS for pelvic girdle pain (<https://pubmed.ncbi.nlm.nih.gov/22722614/>), even though TENS was (and still is) widely used in physiotherapy practice in Sweden. We wanted to compare TENS to the best-evidence treatment available, which was acupuncture, with the aim to investigate if TENS was as least as good as acupuncture for pain relief. This makes our study similar to a non-inferiority trial as we didn't try to test if TENS was superior to acupuncture. In the STRICTA checklist, paragraph 2b refers to specific objectives or hypotheses, where we see the aim of the trial as our objective and therefore have chosen not to add a specific hypothesis in the manuscript.

Can you explain in more detail the location of electrodes in TENS group? I could not find this information on methods.

Reply: We agree that this could be clarified, and we added the following information in the manuscript (pg 10, line 175-76): "...either uni- or bilaterally over the sacroiliac joints and gluteal muscles for dorsal pelvic girdle pain or in the groin area for pubic pain."

Page 19. #303, The difference between groups chosen for the sample size calculation (1 point) seems quite small, especially for the PSFS. Is that clinically relevant for this population? Where were the SDs estimated from? Can you please provide references?

Reply: Unfortunately, we are not sure about which paragraph in the manuscript these questions refer to, but PSFS was not used for the sample size calculation. There is no MCID at present for PSFS in a pregnant population. On page 19 line 327, we refer to Horn (<https://pubmed.ncbi.nlm.nih.gov/22031594/>) that reports small change: 0.8; medium change: 3.2; large change: 4.3 for PSFS in mechanical low back pain, which was chosen as the best alternative. We have added this reference also on page 20, line 352. Regarding the comment about SD:s, we are sorry, but we do not understand this question.

Page 30. Figure 2.

Despite the table showed the effects sizes of both interventions. It is clear the majority of effects sizes is in the group Acupuncture. Do you have any reasons for that? The acupoints selected possible can interfere into the results expectations?

Reply: As this comment refer to Figure 2, (side-effects of the interventions) we have interpreted that the question refer to this and not effect sizes. We do not have any specific reason for that more participants in the acupuncture group experienced mild side effects of the treatment. It would only be speculations, so we chose not to address it further in the manuscript. Maybe the women in the TENS group experienced being in control and adjusted the intensity of the stimulation so it felt comfortable or that acupuncture is perceived as a more invasive procedure that affected the participants in various ways.

In addition, any recommendations were delivered due to the effects sides in pregnancy?

Reply: This is a relevant question and we have added this information in the manuscript (pg 8-9 lines 172-74) "The participants got written information about acupuncture at the start of treatment and were instructed to report any perceived side effects to the treating physiotherapist."

Reviewer: 3

Competing interests of Reviewer: None

Reviewer: 1

Competing interests of Reviewer: I do not have competing interests.